# How to Overcome a Snail? Identification of Putative Neurotoxins of Snail-Feeding Firefly Larvae (Coleoptera: Lampyridae, *Lampyris noctiluca*)

**DOI:** 10.3390/toxins16060272

**Published:** 2024-06-14

**Authors:** Jonas Krämer, Patrick Hölker, Reinhard Predel

**Affiliations:** 1Institute of Zoology, University of Cologne, Zuelpicher Strasse 47b, 50674 Cologne, Germany; 2Institute for Insect Biotechnology, Justus-Liebig-University of Giessen, Heinrich-Buff-Ring 26-32, 35392 Giessen, Germany

**Keywords:** venom, neurotoxin, extra-oral digestion, neglected venomous animals, lampyrid beetle larvae

## Abstract

The larvae of some lampyrid beetles are highly specialized predators of snails. They have been observed to climb on the shells of their prey and use this exposed position to bite and inject secretions potentially originating from the midgut. Besides serving the purpose of extra-oral digestion (EOD), injected compounds also seem to have a paralyzing effect. Up to now, the toxins causing this paralyzing activity have not been identified. In the current study, we provide a first compositional analysis of the midgut secretion from lampyrid larvae, with a focus on identifying putative neurotoxins causing the observed paralyzing effect. For this purpose, we utilized a combined proteo-transcriptomic approach to characterize the compounds present in the midgut secretion of larval stages of *Lampyris noctiluca*. In terms of the absolute numbers of identified compounds, the midgut secretion is dominated by hydrolyzing enzymes comprising peptidases, carboxylesterases, and glycosidases. However, when considering expression levels, a few rather short cysteine-rich peptides exceed all other compounds. Some of these compounds show moderate similarity to putative neurotoxins identified in the venom of other arthropods and could be responsible for paralyzing effects. In addition to these potential toxins, we provide a list of peptides typical of the midgut secretion of *L. noctiluca*, supplemented by the corresponding precursor sequences.

## 1. Introduction

Lampyrid beetles are famous for their bioluminescence, used mainly by adults as part of their courtship behavior. The adult beetles rarely feed and in many species the mouthparts are reduced in this stage[1]. Hence, the necessary energy for the light spectacle must be acquired by their voracious predatory larvae [2]. In some species like *Lampyris noctiluca*, the larvae are highly adapted for tracking down and subduing snails and also worms [3,4]. To overcome much larger prey, *L. noctiluca* larvae have evolved an interesting strategy involving morphological, behavioral, and chemical adaptations [3,4,5] (Figure 1). First, for tracking their prey, *L. noctiluca* larvae use their maxillary palps equipped with highly sensitive chemoreceptors to detect slime trails that can be several days old. Once encountering a snail, the larvae climb onto the shell and attach to it with a modified pygopodium. Riding on its prey, the *L. noctiluca* larvae finally bite to inject a deadly cocktail [5]. The secretion is released via a narrow opening located just before the tip of the hollow mandibles connected to the oral cavity [6].

Besides the digestive function of this secretion, a paralyzing effect on the prey has also been described by several authors [1,5,7]. As *L. noctiluca* larvae lack distinctive salivary or venom glands associated with their mandibles, the midgut was identified as the most likely production site of paralyzing compounds, presumably neurotoxins [1,7]. Hence, the usage of the toxic secretion by lampyrid larvae could fulfill the dual purpose of extra-oral digestion (EOD) and envenomation. In the extreme species-rich order of Coleoptera, the usage of prey-paralyzing toxins has so far only been assumed for a few groups [8], among them lampyrid larvae. However, the composition of the secretion utilized by these larvae for predation has not yet been elucidated. Like many other venomous arthropods that pose no threat to humans, lampyrid larvae have not been the focus of venom research [9,10]. Additionally, their small size has hampered the compositional analysis of their toxic secretion, as traditional approaches require large quantities of material. With the advent of modern omics approaches, the focus of venom research has recently been expanded toward species that provide only minute amounts of venom [11]. Besides new insights into venom evolution, sampling across a wider taxonomic range has the potential of identifying bioactive compounds with a high degree of novelty [12]. This was demonstrated by studies on the venom composition of several other groups previously neglected in venom research [13,14,15,16,17]. Toxins from animal venoms are known to target a plethora of physiological processes and hence are suitable candidates for new lead compounds [18,19]. So far, at least eleven approved drugs have been developed based on compounds discovered in animal venoms [20]. In addition, several compounds with promising activities against diseases like epilepsy, chronic pain, autoimmune diseases, stroke, and diabetes [21,22,23,24,25] emphasize the potential of animal venoms in this regard.

In the present study, we apply a proteo-transcriptomic approach to analyze the composition of secretions extracted from the gastrointestinal tract (putatively originating from the midgut and therefore later on referred to as midgut secretion) of lampyrid larvae of the species *L*. *noctiluca*. This enabled the discovery of putative neurotoxins that might cause the previously described paralyzing effect in their prey.

## 2. Results

To investigate the composition of putative midgut secretions from larvae of *L*. *noctiluca*, we established a reliable extraction procedure. For this purpose, the animals were immobilized utilizing adhesive strips, which also ensured accessibility to the retractable larval head (Figure 2A). Applying electricity to the anterior part of the larval abdomen reliably triggered the release of a transparent brownish secretion, either from the mandibles or the pharynx opening (Figure 2B). This was collected with glass capillaries (Figure 2C).

MALDI-TOF mass spectra of repeatedly released midgut secretions from four larvae of *L*. *noctiluca* showed that the midgut secretion of these larvae is subject to some degree of variation (Appendix A). On the one hand, deviating signals occurred for samples extracted from freshly collected individuals (observed for two of the specimens). On the other hand, for one of the specimens (larva 4), the spectra of the midgut secretions showed several additional signals and strong differences in the relative intensities of the measured signals compared to the other specimens (Appendix A). Nevertheless, based on mass spectra obtained for midgut secretions from several repeated extractions, well-reproducible signals were identified regardless of the discharge site of the secretion (mouth opening or tip of mandibles). Most of the reproducibly detected compounds were observed in the mass range of *m*/*z* 2000–4500 (Figure 3C, Appendix A).

The transcriptome sequencing of midgut tissue from a single specimen produced 35,583,691 raw reads, which were assembled into 135,615 contigs. The assessment of transcriptome completeness with BUSCO yielded 94.8% complete BUSCOs and 1.5% fragmented BUSCOs. Matching the proteomic (one top-down and one bottom-up analysis) data against this reference transcriptome yielded 286 matches after quality filtering and removal of precursors without a signal peptide/stop codon (Appendix A). The functional annotation performed based on BLAST and InterProScan results indicates that approximately half of the identified precursors are potentially involved in housekeeping (Appendix A). Figure 3A,B provide an overview of those functional classes for which a direct involvement in envenomation/EOD of lampyrid prey can be assumed, and Table 1 lists the precursors of those toxins potentially causing the paralytic effects on prey. In terms of absolute numbers, most of the precursors identified in the midgut secretion of *L*. *noctiluca* were annotated as different types of hydrolases, comprising peptidases, carboxylesterases, and glycosidases. Interestingly, when considering the expression level, it is the class of putative neurotoxins clearly exceeding all others. The classification as a putative neurotoxin is based either on similarity to compounds from the Tox-Prot database or on InterProScan results. All putative neurotoxins contain at least four cysteines and range in size between 3 and 12 kDa. High expression levels were mainly observed for three highly similar peptides, U-Lampyristoxin-Ln1a-c, with identical cysteine patterns. The classification as neurotoxin is based on a BLAST hit for U-Lampyristoxin-Ln1a with a putative potassium channel blocker from scorpion venom (Figure 4), although the similarity is relatively low and the cysteine pattern is not identical. The structure prediction of Alphafold 2 performed for this compound indicates the presence of a single disulfide bridge and an alpha-helical conformation of the C-terminal part of the peptide (Appendix A). The predicted model, though, exhibits a high uncertainty of 26.98% (Appendix A).

The high expression levels of Lampyristoxin-Ln1a-c are also reflected in their high relative ion intensities in the MALDI TOF mass spectra of midgut secretions. Most of the well-reproducible signals could be allocated to the Lampyristoxin-Ln1a-c (Figure 3C) or intermediate products of their precursor processing. Regarding precursor processing, the pro-peptide of U-Lampyristoxin-Ln1a is cleaved at a quadruplet motif [26], whereas in the case of Lampyristoxin-Ln1b-c, a single arginine is likely functioning as a monobasic cleavage signal. Besides Lampyristoxin-Ln1, the only other relatively highly expressed putative neurotoxin is Lampyristoxin-Ln2. The ion signal of the predicted mature peptide of this precursor was repeatedly observed in the MALDI-TOF mass spectra of midgut secretions (Figure 3C). Based on the InterProScan results, this compound was classified into the scorpion-toxin-like superfamily, though it also exhibits similarity to defensins. Based on the structure prediction of Alphafold 2, a cysteine-stabilized α/β domain was assumed for this peptide (Appendix A). U-Lampyristoxin-Ln5a and b exhibit similarity to neurotoxin-like cysteine-rich peptides described for a non-venomous springtail and, at least for U-Lampyristoxin-Ln5b, a similarity to defensins is indicated by the BLAST hit from the Metazoa database (Appendix A).

The longest putative neurotoxins are U-Lampyristoxin-Ln3 and U-Lampyristoxin-Ln4, which exhibit similarity to venom compounds identified for different species of centipedes. The main difference between these toxins is their cysteine content, which is, with 12 cysteines, very high in U-Lampyristoxin-Ln3 and, with four cysteines, relatively low in U-Lampyristoxin-Ln4. For the latter, cleavage of a propeptide at a quadruplet motif is assumed, and InterProScan also indicates a moderate similarity to the superfamily of odorant-binding proteins. Compounds classified into this superfamily were discovered in high abundance in the midgut secretion, and at least some of these exhibit relatively high expression levels (Figure 3A, Appendix A). All described putative neurotoxins show only moderate similarity to previously described venom compounds, and there is a smooth transition toward compounds classified as novel. Of these novel compounds, only two are included in Table 1, for which corresponding signals in the MALDI-TOF mass spectra of midgut secretions are present. Both Novel-Lampyris-Venom-Compounds 1 and 2 (NLVC 1 and 2) contain six cysteines, and the mature peptides are cleaved directly C-terminal to the signal peptide.

## 3. Discussion

The current study provides the first compositional analysis of the midgut secretion utilized by lampyrid beetle larvae for EOD and, potentially, for paralyzing their prey. EOD is a feeding mode that evolved in many arthropod groups and enables feeding on prey that is much larger than the individual predator [27]. The original source of digestive enzymes in arthropods is the midgut [27]. In some groups like predatory heteropterans, digestive enzymes were recruited into the salivary/venom glands, and their venom is adapted for the dual purpose of paralyzing and liquifying the prey [28]. Other groups with distinctive venom delivery systems, e.g., spiders, still mainly rely on midgut secretions for the purpose of EOD [29]. In the case of predatory beetles or beetle larvae that rely on EOD for feeding, midgut secretions are regurgitated to liquefy prey [27]. At least for some coleopteran subgroups, potential neurotoxic effects on their prey were assumed in the context of EOD [8]. Among these, the lampyrid beetle larvae investigated in this study are extraordinary with their strategy to ride their much larger prey and incapacitate it by several consecutive injections of gastrointestinal secretion [5]. As distinct salivary or venom glands are absent in these larvae, the midgut was also assumed as the production site of potential neurotoxins causing the observed effects on snails [7]. Depending on the preferred definition, the toxic secretions used by *L. noctiluca* larvae to subdue prey, and, more generally, secretions used by arthropods for EOD, may or may not be described as venom (e.g., [30,31]). EOD secretions can be defined as venom based on their injection into the internal milieu of another organism by inflicting a mechanical injury (e.g., [30]). The main difference, e.g., compared to predatory heteropterans, which are always classified as venomous, is that in *L. noctiluca* and other predatory coleopterans (larvae and imagines), the toxins for prey digestion are not produced in specialized venom glands, but solely in the midgut. Based on this difference, conventional venom definitions might exclude EOD secretions produced in the midgut from being classified as venom (e.g., [31]). To avoid a debate in this regard, we decided in favor of a neutral description in our manuscript. In the current work, we identified potential neurotoxins in the midgut secretion of the larva of *L*. *noctiluca*. Some of the identified compounds are highly expressed by the glandular tissue of the midgut, indicating a high energy investment. These findings support the hypothesis that compounds causing the observed paralyzing effects on the prey of *L. noctiluca* larvae might indeed be produced in the midgut.

The identified putative neurotoxins exhibit only moderate similarity to annotated neurotoxins from the Tox-Prot database. Hence, assumptions about their targets are speculative, especially as single amino acid substitutions can change the target selectivity of a neurotoxin [32]. The highly expressed U-Lampyristoxin-Ln1a-c and U-Lampyristoxin-Ln2 show similarity to a scorpion toxin family mainly comprising potassium channel blockers. Toxins in this family, in turn, exhibit similarity to defensins, which are the most likely progenitors of scorpion potassium channel blockers [33]. The toxin signature identified for scorpion potassium channel blockers [33], though, is missing in both U-Lampyristoxins-Ln1 and 2. In the case of U-Lampyristoxin-Ln1, the cysteine pattern also strongly deviates from the BLAST hit, and especially, the two C-terminal cysteines are uncommon. Therefore, it is questionable if the cysteine pattern of U-Lampyristoxin-Ln1a-c fits the general structural motifs like the ICK described for animal toxins, e.g., from spiders [34,35]. The structure prediction performed with Alphafold 2 for U-Lampyristoxin-Ln1a also provided a model with high uncertainty (Appendix A). For this sequence, Alphafold 2 was unable to predict the correct disulfide bridges, as only a single disulfide bridge between the second and fourth cysteine was predicted (Appendix A). However, our proteomic data confirmed the presence of three disulfide bridges for this compound. These findings demonstrate the high degree of sequence deviation of U-Lampyristoxin-Ln1a compared to already described cysteine-stabilized peptides. In contrast, a more precise model was predicted for U-Lampyristoxin-Ln2 with Alphafold 2, indicating a cysteine-stabilized α/β domain. This structural motif is common for the scorpion toxin-like/defensin family, and the Alphafold model is in accordance with the InterProScan prediction for this peptide. U-Lampyristoxin-Ln6 is another putative toxin with similarity to a compound from scorpion venom. In this case, the BLAST hit was annotated as beta toxin, which indicates a potential modulation of the inactivation process of sodium channels. Surprisingly, U-Lampyristoxin-Ln7 is the only putative neurotoxin with sequence similarity to a neurotoxin identified in the venom of another insect (U-Asilidin(12)-Dg3b, *Dolopus genitalis*). A possible explanation for this is that toxins from insect venom are still underrepresented in the Tox-Prot database, as many groups remain poorly investigated [10]. Regarding precursor processing, a distinctive feature of some of the putative neurotoxins (U-Lampyristoxin-Ln1b + c and U-Lampyristoxin-Ln2) is the cleavage of the pro-peptide at a single arginine. For most described venom compounds, pro-peptide cleavage occurs either at dibasic or quadruplet motifs [26,34]. The cleavage at a single arginine is supported by the presence of prominent ion signals in MALDI-TOF mass spectra that match with the masses of the predicted mature peptides and by the Q-Exactive fragment data. Other abundant compounds contributing to the midgut composition of *L*. *noctiluca* are classified as odorant-binding proteins. The original function of odorant-binding proteins is to bind hydrophobic signaling compounds, like pheromones, and enable an interaction with their corresponding receptors [36]. Venom compounds with similarity to odorant-binding proteins have also been discovered in other insect venoms [37,38,39]. For predatory heteropterans, a function in detecting and sensing dietary materials was assumed [38], but their function in venom remains largely unexplored [40]. Of the two relatively large toxins, U-Lampyristoxin-Ln3 and 4, with similarity to venom compounds of centipedes, at least U-Lampyristoxin-Ln4 also shows similarity to the odorant-binding protein family. This might be an indication that, at least for some of the odorant-binding proteins discovered in the midgut secretion of *L*. *noctiluca*, their original function might have been modified. In general, the sequences of peptide/protein neurotoxins that have been identified in animal venoms to date are highly divergent and often contain only a few or very short conserved sequence regions [18,33,34]. One characteristic shared by most animal neurotoxins is their cysteine-stabilized scaffold, which is essential for proper functioning and high stability [18]. Also, for the potential neurotoxins identified in the *L. noctiluca* midgut secretion, no conserved sequence domains could be identified. Only the isoforms of Lampyristoxin-Ln1 and Lampyristoxin-Ln5 show a high degree of sequence similarity. Apart from the compounds with moderate sequence similarity to animal neurotoxins, the midgut compounds with no relevant BLAST hits also comprise several cysteine-rich peptides. These ‘novel’ compounds are additional candidates that might exhibit neurotoxic activities, and at least for NLVC 1 and NLVC 2, their potential presence in midgut samples was repeatedly demonstrated with MALDI-TOF MS.

Apart from the potential paralysis of prey, *L. noctiluca* larvae inject their midgut secretion for EOD. This purpose is fulfilled by the various hydrolyzing enzymes that, in terms of abundance, dominate the midgut secretion of *L*. *noctiluca*. The identified enzyme classes such as peptidases and glucosidases are frequently found in insect midguts, and their specific functions have been discussed in detail, e.g., in Terra and Ferreira [41]. Regarding the potential neurotoxic effect of gastrointestinal secretion, Schwalb assumed that only the initial bites of *L*. *noctiluca* cause the neurotoxic effects and concluded that venom glands in the pharynx region must be responsible [5]. An alternative explanation is that neurotoxin production might only occur in a restricted section of the midgut. This is supported by the fact that the composition of cell types in the glandular tissue of the insect midgut is also inconsistent and alternates from the anterior to the posterior section [42]. In this regard, it was already shown that the anterior and posterior sections of the coleopteran midgut exhibit differences in pH and contain different enzymes [43]. Moreover, the composition of the midgut secretion is subject to dynamic changes induced by complex hormonal regulation [44]. Nevertheless, in the MALDI-TOF mass spectra of midgut secretions from different specimens of *L*. *noctiluca*, we identified several well-reproducible ion signals and assigned most of these to some of the highly expressed midgut compounds.

To confirm whether the midgut secretions of *L. noctiluca* larvae cause prey paralysis, in vitro and in vivo assays are necessary. In the current work, we identified several candidate compounds with moderate similarity to arthropod neurotoxins. Future work should focus on obtaining the individual midgut compounds, e.g., by recombinant production, and using them for activity tests on potential prey or at the cellular level to elucidate the mode of action of the potential toxins. Finally, comparative transcriptomics or qPCR of different midgut sections can be used to examine the proposed differential expression of *L. noctiluca* midgut compounds.

## 4. Materials and Methods

### 4.1. Collecting and Rearing of L. noctiluca Larvae

Last-stage larval specimens of *L*. *noctiluca* were manually collected in Cologne, North Rhine-Westphalia, Germany (50°57′ N 6°52′ E). The larvae were found in the daytime, crawling up tree stems. For rearing, the animals were placed in plastic boxes (15 × 15 × 20 cm) equipped with coco substrate and bark pieces, at a constant temperature of 21 °C. To ensure sufficient ventilation, the lids of the boxes were partially punched out and covered with fine wire mesh. Humidity was adjusted by watering some of the substrate once a week. The larvae were fed with earthworms (*Lumbricus* or *Dendrobena*).

### 4.2. Extraction of Midgut Secretion for Proteomics

The release of midgut secretion was triggered by the application of electricity at the first abdominal segments using a Promed tens device (Promed GmbH, Farchant, Germany; pulse width 200–250 ms, pulse rate 60–130 Hz, voltage between 8 and 12 V) coupled with a self-made electrical forceps. To ensure sufficient stimulation, conductive fluid was applied. Emerging secretions were collected with pulled glass capillaries and then diluted either with ultrapure water or an ethanol/TFA mixture (35%/0.1%).

### 4.3. MALDI-TOF Mass Spectrometry

Diluted secretion (0.3 µL) was spotted on a MALDI-TOF sample plate and mixed on-plate with an equal volume of 10 mg/mL 2.5-dihydroxybenzoic acid (Sigma Aldrich, Steinheim, Germany) matrix, dissolved in 50% acetonitrile/0.05% TFA. For an optimal crystallization of matrix salts, the samples were blow-dried with a hairdryer. Mass spectra of midgut secretions were generated on a Bruker ultrafleXtreme TOF/TOF mass spectrometer (Bruker Daltonik GmbH, Bremen, Germany) in reflectron positive mode. The spectra were generated in mass ranges of *m*/*z* 800 to 4500 and 3000 to 10,000. Only ion signals in the lower mass range could be assigned to data from transcriptomic and Q-Exactive MS data, therefore the MALDI-TOF MS data presented here are restricted to this mass range. For external calibration, a mixture containing proctolin ([M + H]^+^, 649.3), Drm-sNPF-212-19, ([M + H]^+^, 974.5), Pea-FMRFa-12 ([M + H]^+^, 1009.5), Lom-PVK ([M + H]^+^, 1104.6), Mas-allatotropin ([M + H]^+^, 1486.7), Drm-IPNa ([M + H]^+^, 1653.9), Pea-SKN ([M + H]_+_, 2010.9), and glucagon ([M + H]^+^, 3481.6) was used. Ion signals were identified by using the peak detection algorithm, SNAP, from the flexAnalysis 3.4 software package. In addition, each spectrum was manually checked to ensure that the monoisotopic peaks were correctly identified.

### 4.4. Top-Down and Bottom-Up Proteomics

The proteome of midgut secretions of larval stages of *L. noctiluca* was analyzed using both top-down and bottom-up strategies as described in [45]. Briefly, protein concentrations were determined with a Direct Detect Spectrometer (Merck, Darmstadt, Germany). For both the top-down and bottom-up approach, urea buffer (8M urea/50 mM triethylammonium bicarbonate buffer, final concentration of urea (7M)) was used for the denaturation of proteins prior to the reduction and alkylation with dithiothreitol and chloracetamide. For the bottom-up approach, 25 µg of the sample was digested with trypsin (Sigma-Aldrich, St. Louis, MO, USA), applying an enzyme/substrate ratio of 1/75. For the desalting and removal of urea, a poly (styrene divinylbenzene) reversed-phase (RP) StageTip purification protocol was used. The final proteomic analysis was performed with a Q-Exactive Plus (Thermo Fisher Scientific) mass spectrometer coupled to an EASY nanoLC 1000 UPLC system (Thermo Fisher Scientific, Bremen, Germany). For the UPLC, inhouse packed RPC18 columns with a length of 50 cm were used (fused silica tube with ID 50 µm +/− 3 µm, OD 150 µm; Reprosil 1.9 µm, pore diameter 60 A°; Dr. Maisch GmbH, Ammerbuch-Entringen, Germany). The UPLC upstream separation of midgut compounds/tryptic peptides was performed with a binary buffer system (A: 0.1% formic acid (FA), B: 80% acetonitrile, 0.1% FA), linear gradient from 2 to 62% in 110 min, 62–75% in 30 min, and final washing from 75 to 95% in 6 min (flow rate 250 nL/min). Re-equilibration was performed with 4% B for 4 min. To obtain fragment spectra with the Q-Exactive Plus, HCD fragmentations were performed for the 10 most abundant ion signals from each survey scan in a mass range of *m*/*z* 300–3000. The resolution for full MS1 acquisition was set to 70,000 with an automatic gain control target (AGC target) at 3 × 10^6^ and a maximum injection time of 80 ms. The run was performed at a resolution of 35,000, AGC target at 3 × 10^6^, a maximum injection time of 240 ms, and 28 eV normalized collision energy; dynamic exclusion was set to 25 s.

### 4.5. RNA Extraction and Sequencing, and Transcriptome Assembly

For RNA extraction, the midgut of one larval specimen of *L. noctiluca* was transferred to TRIzol (Thermo Fisher Scientific, Darmstadt, Germany) and homogenized using scissors, the pistil, and a sonic finger. RNA was extracted according to the standard TRIzol user guide, and, subsequently, the RNA concentration was measured with Qubit (Thermo Fisher Scientific, Waltham, MA, USA). Library preparation was achieved for 1 µg of total RNA using the Illumina TruSeq Stranded RNA Sample Preparation Kit. The libraries were validated and quantified with an Agilent 2100 Bioanalyzer, and paired-end sequencing was performed with an Illumina TruSeq PE Cluster Kit v3 and an Illumina TruSeq SBS Kit v3—HS on an Illumina HiSeq 4000 sequencer. For quality trimming and adapter removal of raw data, a Trimmomatic 0.3.2 [46] was used, and transcriptome de novo assembly was performed with Trinity v2.8.5 [47], applying default settings. Finally, the completeness of transcriptomic data was assessed with BUSCO [48] and the data submitted to the National Center for Biotechnology Information (NCBI), (Bioproject: PRJNA1120466, Biosample: SAMN41694318).

### 4.6. Identification and Annotation of Midgut Peptides/Proteins

For peptide identification/assignment to precursors, the fragment spectra generated by top-down and bottom-up proteomics were matched against the midgut transcriptome utilizing the software PEAKS 10 (PEAKS Studio 10; BSI, Toronto, ON, Canada). For the recognized precursors, potential open reading frames were identified. Precursor integrity was then assessed by examining for the presence of a stop codon and by running SignalP 6.0 [49] to check for the presence of a signal peptide. For functional annotation, InterProScan was used to assign protein families and GO terms. In addition, BLAST searches against the Metazoa database (search term: ‘(taxonomy_id:33208)’) and Tox-Prot database (search term ‘(keyword:KW-0800) AND (taxonomy_id:33208)’)from UniProt [50] were used as the basis for annotation. The E-value thresholds for the BLAST searches were set to 1 × 10^−5^ in the case of the Metazoa database and to 1 × 10^−2^ in the case of the Tox-Prot database. Among the ten BLAST hits with the lowest E-values, the one with the highest accordance between alignment length, query length, and BLAST hit length was selected as the best hit. Finally, the identified midgut precursors were filtered. Only complete precursors with a signal peptide and stop codon were kept, for which the coverage between transcriptomic and proteomic data was at least 7%, and the false discovery rate (−10lg(P)) of the PEAKS run was at least 30. To examine if the processed peptides with a mass below 4500 Da can be allocated to the precursors identified by the proteo-transcriptomic analysis, theoretical masses calculated for each of the peptides (predicted from the identified precursors) were searched against a list of ion signals from the MALDI-TOF mass spectra of the midgut secretion. For some of the identified midgut compounds, we utilized Alphafold 2 [51,52] for structure prediction with default settings. Based on the output of Alphafold 2, we also assessed the disulfide bridges of these compounds by calculating the distance between the sulfur atoms of the cysteines.

## Figures and Tables

**Figure 1 toxins-16-00272-f001:**
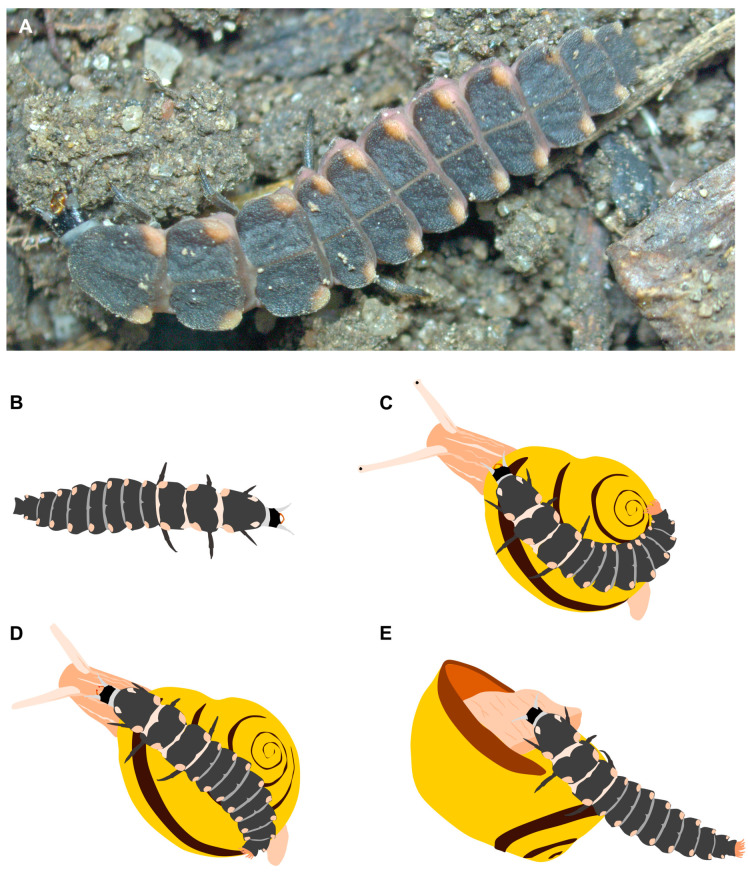
Habitus and feeding strategy of larval stages of *L*. *noctiluca*. (**A**) Photo of a foraging *L*. *noctiluca* larva. (**B**–**E**) Illustrations showing the feeding strategy: First, a larva detects the slime trails of snails with its maxillary palps. (**B**). Then, the larva climbs onto the shell of its prey, attaches itself with a modified pygopodium (**C**), and injects a toxic secretion into its prey’s neck via hollow mandibles (**D**). When the snail is paralyzed, the larva feeds by utilizing extra-oral digestion (**E**).

**Figure 2 toxins-16-00272-f002:**
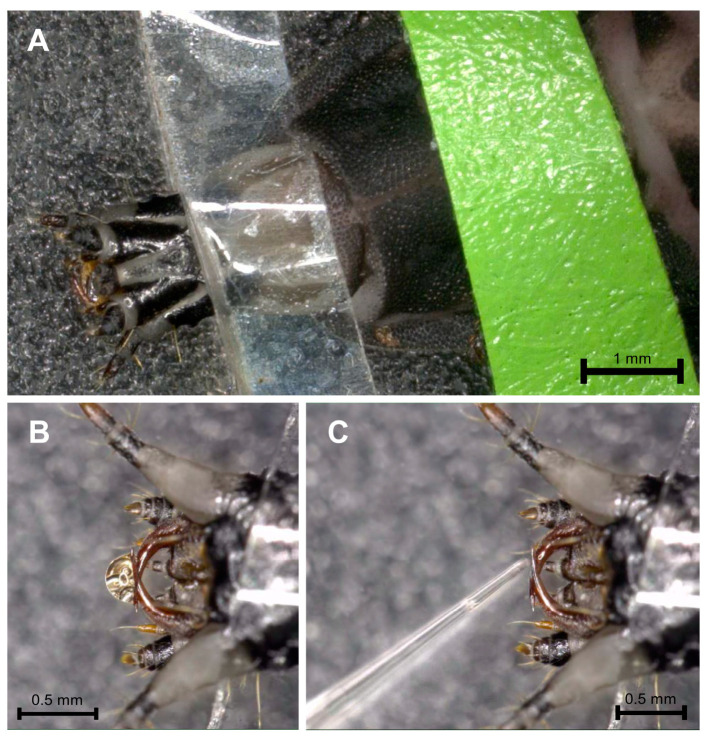
Procedure for extracting midgut secretions from larvae of *L. noctiluca*. The larvae were immobilized with adhesive strips (**A**), and the release of the midgut secretion was triggered by the application of electricity (**B**). The secretion was collected with pulled capillaries (**C**).

**Figure 3 toxins-16-00272-f003:**
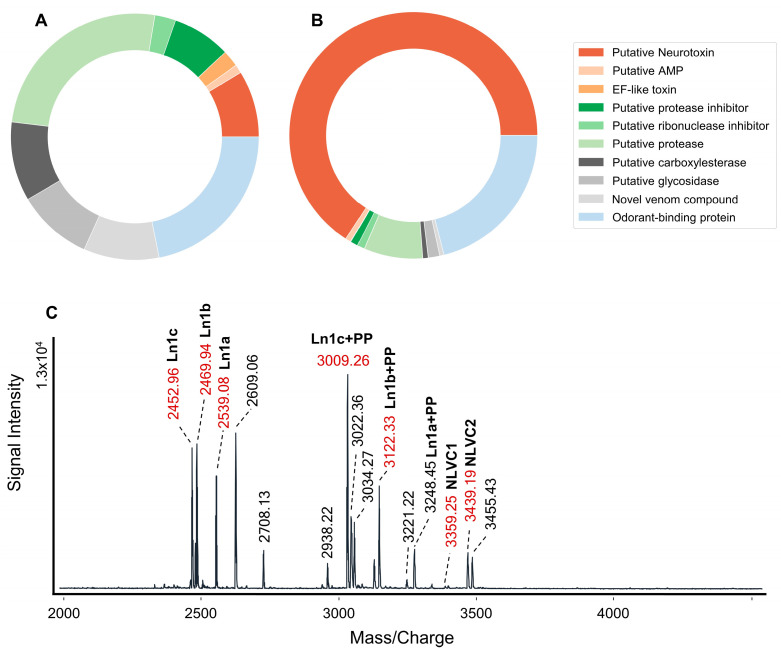
Overview of compound classes identified in midgut secretions from lampyrid beetle larvae of *L. noctiluca*. Compound classes with a putative housekeeping function are not included: (**A**) Absolute numbers of precursors assigned to the identified substance classes. (**B**) The summed expression level of the identified compound classes [tpm]. (**C**) MALDI-TOF mass spectrum (*m*/*z* 2000 to 4500) of midgut secretion extracted from an individual larva. Signals with a mass match to identified midgut compounds are labeled accordingly. Additionally, the most reproducible signals are highlighted in red (occurrence in at least half of the 13 spectra from 4 individuals; larva 2: five extractions; larva 3: three extractions; larva 4: three extractions; larva 1: single extraction only; subsequently this specimen was used for RNA extraction). PP = propeptide.

**Figure 4 toxins-16-00272-f004:**
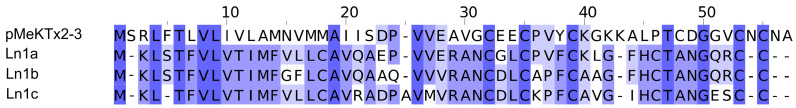
Alignment of all isoforms of the highly expressed U-Lampyristoxins-Ln1 with the corresponding BLASThit, a putative potassium channel blocker identified in the venom of *Mesobuthus eupeus*. The coloration indicates percentage of identity.

**Table 1 toxins-16-00272-t001:** Precursors of putative neurotoxins identified in midgut secretions of *L. noctiluca*. Precursors were identified by a combined proteomic and transcriptomic approach. Grey, signal peptide; blue, potential bioactive peptide; yellow, cysteine (half of the disulfide bond); red, potential cleavage site. Black underlined, confirmed by MSMS; red underlined, confirmed only by MSMS digested samples; dashed line, mass match in MALDI-TOF MS. The column Orbitrap MS indicates which proteomic approach yielded fragment spectra that matched to the reference transcriptome. MALDI-TOF MS: Confirmation by mass match. Confirmed PTMs only comprise disulfide bridges (C-C). Potential C-terminal amidations (Ln2, 4) are not corroborated by our data.

Name	BLAST/InterProScan	Expression Level [tpm]	PTM	Predicted Mass [M + H^+^]	OrbitrapMS	MALDI-TOF MS
U-Lampyristoxin-Ln1a	Potassium channel blocker pMeKTx2-3 (*Mesobuthus eupeus*), 38%, Acc: A0A088D9Q6	18,044	C-C	2539.01/3248.4	Bottom-up + Top-down	+
	MKLSTFVLVTIMFVLLCAVQA ** EPVV ** ** ER ** ** ANCGLCPVFCKLGFHCTANGQRCC **					
U-Lampyristoxin-Ln1b	-	13,146	C-C	2469.89/3122.27	Bottom-up + Top-down	+
	MKLSTFVLVTIMFGFLCAVQA ** AQVVV ** ** R ** ** ANCDLCAPFCAAGFHCTANGQRCC **					
U-Lampyristoxin-Ln1c	-	8310	C-C	2452.91/3009.22	Top-down	+
	MKLTFVLVTIMFVLLCAVRA **DP** ** AVMV ** ** R ** ** ANCDLCKPFCAVGIHCTANGESC ** ** C **					
U-Lampyristoxin-Ln2	Scorpion toxin-like/defensin (InterProScan)	7326	C-C	4378.8	Bottom-up + Top-down	+
	MNRSIFILLLVISVLFAAVVAVPIHEKEELPHLMLYT**RAVSCKAVSSRPNDPSSYNEACNAHCILNGNRGGVCGSGTCICLG**					
U-Lampyristoxin-Ln3	U-scoloptoxin(19)-Sm1a (*Scolopendra morsitans*), 33%, P0DQE9	2	C-C	10,661.77	Bottom-up	-
	MFSSLLLICLLPILVLG ** TGTSGPVDFHPEEPCNR ** ** AGGQCIKRDECPVHIEDIYLNLCPQQQSQGAECCHGISTKEYRCK KFGGECFREGSK CPDNLK RPQATDCPAGKFCCVLI **					
U-Lampyristoxin-Ln4	U-scoloptoxin(17)-Er3a (*Ethmostigmus rubripes*), 35%, P0DQE6, Odorant-binding protein (InterProScan)	56	C-C	12,803.47	Bottom-up	-
	MKWLLCFVIACALRVYS **KRINVGALVP** ** ER ** ** ECLKDYRDNFPKIIYALYSISPSNDEVVGEYFICTLKKRQILEDNGEINPEKIYKYWVEVYQTTIISPSEEKEISDAAEECAKLKDDKMAFLALKIKNCILEGAHKLPFVG **					
U-Lampyristoxin-Ln5a	Long neurotoxin OH-34 (*Orchesella cincta*), 43%, A0A1D2NF32	6	C-C	4839.16	Bottom-up	-
	MVILAIFGRVDA ** ADVSLGCTLSCSIWNACRVKAALSGNLESCGPQPGGCRCTQFAWER **					
U-Lampyristoxin-Ln5b	Long neurotoxin OH-34 (*Orchesella cincta*), 36%, A0A1D2NF32	18	C-C	5055.25	Top-down	-
	MKNIVLLSVLAMVILAIFGRVDS ** ADISWGCTLSCSIWNACRVKAALSGNLKSCDPQPEGCRCTQFAWER **					
U-Lampyristoxin-Ln6	Putative beta-like toxin Tx770 (*Buthus occitanus Israelis*), 29%, B8XH02	19	C-C	4656.00	Bottom-up	-
	MNRTLVIFLVFIFGFVIAESMV ** VQGGDRYKYCRIAQCKIDCVFQNHIDGFCKNNQCVCTDYN **					
U-Lampyristoxin-Ln7	U-Asilidin(12)-Dg3b (*Dolopus genitalis*), 30%, A0A3G5BIB1	76	C-C	6288.81	Bottom-up	-
	MVRVVIYTTILALMLFNVMA ** GPLLNEDQAQLIRHKRASCSSVTTNGDSRGGWANEGCRAYCVMSGYRTGLCSQGTCACR **					
NLVC 1	-	1	C-C	3359.22	Bottom-up	+
	MKLIIFLLVVCMVFAVPISS ** YCYFCADQCNPGETTYPDSDCPPGK ** ** VSCCKA **					
NLVC 2	-	104	C-C	3439.38	Bottom-up	+
	MKTFLVVLLITILYMSLSVDA ** DCGERCQFMPCRTGY ** ** TGVPERCPGGGIR ** ** CCPP **					

## Data Availability

Transcriptomic data generated in this study were submitted to NCBI [53] (Bioproject: PRJNA1120466, Biosample: SAMN41694318). Transcriptome raw data were submitted to the Sequence Read Archive (Accession number: SRR29299461) and the generated assembly was submitted to the Transcriptome Shotgun Assembly database (Accession number: GKVN00000000). The Orbitrap raw data generated for this project were submitted to the PRIDE database [54] (Accession number: PXD052900).

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
