# Peer review of "How to Overcome a Snail? Identification of Putative Neurotoxins of Snail-Feeding Firefly Larvae (Coleoptera: Lampyridae, Lampyris noctiluca)"

_toxins, 2024, doi:10.3390/toxins16060272_

Round 1

Reviewer 1 Report

Comments and Suggestions for Authors

The manuscript title is creative and catching, the manuscript is quite interesting, and very well conducted and allowed the identification of some Coleopteran molecules involved in mollusk paralysis. The pattern of cysteine residues present on those peptides were identified and blast analysis were performed in order to classify these molecules. The manuscript is clear and well written, the methodology is informative. However, some points should be addressed.

1-      The authors did not mention the deposit of the transcriptomic or proteomic data on databases such as GenBank  or ProteomeXchange. This is crucial.

2-      Page 2 lines 46 and 47: why the authors affirm that the use of toxic secretion is an intermediate stage between EOD and envenomation? Some studies on Arthropoda digestion even in groups basal along evolution present both EOD and envenomation. Protein phylogenetic studies suggests that venom proteins are the result of gene duplication and evolution of the digestive ones. Even in snakes, there are some literature about that. This sentence might be removed or re-written.

3-      Line 109 according to IUBMB enzymes involved in the catalysis of peptide bond should be named as peptidase. Protease is an antique nomenclature.

4-      Line 118: There is a very good review on peptide containing Cys motifs structure (Pineda et al., 2020)  https://doi.org/10.1073/pnas.1914536117. There are some software also capable of domain search and classification that should help to propose those peptides functions.

5-      Lines 194-195 Blast analysis of peptides and proteins, which exhibits C patterns, is quite unproductive since only the 3 D structure are maintained and they present distinct specificity and properties. Molecular modeling at alpha-fold 2, which is easily performed, could be informative in order to structurally classify the lampyrtoxins identified.

6-      Lines 205-206: Was the secretion tested in other animals like insect larvae or Nematoda?

7-      Lines 205-206: Would it be possible to perform a qPCR in distinct midgut sections evidencing the expression of the molecules at the anterior region?

8-      Line 234: There is a lot of reference of the different cell morphology, physiology and distinct secretion along the midgut of other Coleoptera and even other insect species. Even pH and the secretion of enzymes involved in digestion are compartmentalized in insect midgut, (see Terra and Ferreira for that). This discussion would enrich the manuscript.

Reviewer 2 Report

Comments and Suggestions for Authors

The authors identified some putative neurotoxins from midgut secretions of lampyrid larvae by an combined proteo-transcriptomic approach, and the results can support the hypothesis that compounds causing the observed paralyzing effects on prey of lampyrid larvae are indeed produced in the midgut. The findings will expand the knowledge on diversity and origin of insect venom.

I suggest the authors to conduct some msinor improvment.

1, The potential causes of divergence in composition between proteome and transcriptome should be discussed.

2, More information on NLVC 1 and NLVC 2 should be described.

3, Are there any conserved domain among similar compounds?

Reviewer 3 Report

Comments and Suggestions for Authors

The manuscript attempts to shed light on the observed neurotoxic affects that Lampyris noctiluca larvae have on their snail prey. The authors utilise a proteo-transcriptomics approach to elucidate compounds in the mid-gut secretions which are claimed to be utilised in their prey immobilisation.

Overall, I think the research has merit, and is warranted, however in my opinion there are some glaring issues with some claims made within the manuscript and its focus.

Major comments:-

English grammar, particularly in the introduction, needs drastically improving. There are unorthodox usages of some words and punctuation such as commas appearing in random places. The manuscript should be looked over in more detail with someone who can correct these grammatical issues.

I think the introduction could benefit from having some pictures of what the adult vs larval stages look like. These are not necessarily well-known species, so just to apply to a wider readership it is always nice to be able to visually see what this species looks like.

I think the venom delivery system of these larvae is one in which puts it into the ‘grey zone’ of the “is it venomous?” debate, which is wholly focused on semantics, philosophy, and evolved ecological purposes. For the most part the authors do well to avoid claiming that these are in fact ‘venomous’, however, I think that this manuscript could benefit greatly by discussing the different definitions of what being ‘venomous’ entails, as there is much debate on this, and many species which skirt the boundaries. This debate would fit particularly well in the discussion, mainly lines 170-190 where the authors have adequately spoken about the recruitment of midgut enzymes being recruited to a venom delivery system in other organisms.

I also don’t think, given the low similarity to a scorpion neurotoxin, that we can postulate that the Lampyristoxin is utilised as a neurotoxin/toxin at all, particularly with seemingly strong claims like in lines 189-190. I think the first hurdle is proving that these compounds are in fact neurotoxic or act as toxins – which without any in vivo or in vitro testing on natural prey items or ion channel targets is difficult to ascertain based off of sequence annotations. I suggest that the authors possibly reduce any instances of strong postulations in this regard and be a little more cautious in their approach, which strangely they have in some instances e.g., the abstract “potential toxins”.

In regard to my previous comment, if the authors aren’t willing to or can’t do any further experiments in isolating/expressing these toxins and testing for their speculated neurotoxicity, then I think it is imperative that the authors discuss where this research should endeavour to go in the future and how they/other research should go about testing these hypotheses.

Minor comments:-

Line 11: I think the sentence ‘Up to now, the composition of 11 the midgut secretion has not been investigated’ can be removed, as you reiterate in the following sentence exactly the same thing.

Line 16: Numerically? This is not the correct word to use here.

Line 18: ‘Outrange’ is also not the correct word to use here. ‘Outnumber’ might be better?

Line 30 – 31: I am not sure this sentence is relevant.

Line 34: As far as I am aware lampyrid is a general term for species within Lampyridae? Using the term lampryid larvae makes it uncertain if these traits are a feature of all larvae of lampyrid beetles, or just a specific group or species such as the one researched in this manuscript. I think this needs to be made more clearer, as it currently seems that all lampyrid larvae have the traits you describe in the introduction.

Line 51: Remove comma between arthropods and that.

Line 52: Remove comma at the end of the sentence.

Line 110: Again, I am not sure ‘outranging’ is the correct term here. Out numbered would be better, I think.

Lines 207 – 208: “This demonstrates that toxins from insect venom are still underrepresented in the Tox-Prot database as many groups remain poorly investigated”. I am not sure that it does. It is entirely possible they could be novel and not match to any known compound. Just because there are little matches has no reflection on the robustness or lack thereof within a database.

Comments on the Quality of English Language

English needs to be improved as there are many punctuation mistakes throughout.

Round 2

Reviewer 3 Report

Comments and Suggestions for Authors

Well done to the authors on updating the manuscript. I really like the addition of figure 1 which now paints a more clearer, visual picture of these larval species.

The manuscript (to me) reads much better and the work will hopefully encourage more research into these seemingly innocuous species. I look forward to seeing where these authors (or other research groups) might discover in regard to these possible neurotoxins.

Just some minor grammatical fixes:-

Line 186: Remove 'like'

Line 200: I am uncertain as to what 'and imagines' refers to?

Line 236: What insect are you referring to?

Line 266: I don't think the acronyms of NLVC1 and 2 are defined prior to this instance?

Comments on the Quality of English Language

English is significantly improved.
